# Molecular Characterization of Heat-Induced HSP11.0 and Master-Regulator HSF from *Cotesia chilonis* and Their Consistent Response to Heat Stress

**DOI:** 10.3390/insects12040322

**Published:** 2021-04-04

**Authors:** Fu-Jing He, Feng Zhu, Ming-Xing Lu, Yu-Zhou Du

**Affiliations:** 1College of Horticulture and Plant Protection & Institute of Applied Entomology, Yangzhou University, Yangzhou 225009, China; hefujing2020@163.com; 2Plant Protection and Quarantine Station of Jiangsu Province, Nanjing 210000, China; changyawen1992@163.com; 3Joint International Research Laboratory of Agriculture and Agri-Product Safety, Yangzhou University, Yangzhou 225009, China

**Keywords:** small heat shock proteins, heat shock factors, *Cotesia chilonis*, temperature stress, gene expression

## Abstract

**Simple Summary:**

Small heat shock proteins (sHSPs) are members of the heat shock protein (HSP) family that play an important role in heat stress, and heat shock factors (HSFs) are transcriptional activators that mainly regulate the expression of HSPs. *Cotesia chilonis*, the major endoparasitoid of *Chilo suppressalis*, widely distributes in China and other Asian regions. Previous studies have shown that *C. chilonis* has a certain thermal tolerance. Here, heat-induced HSP11.0 and master-regulator HSF were cloned and characterized from *C. chilonis*. The transcription patterns of them in response to different temperatures and time course after temperature treatment were analyzed. This study is the first report on the analysis on *hsf* gene of *C. chilonis*. The results of expression patterns will provide new insights into thermoregulation of *C. chilonis* in response to climate change.

**Abstract:**

Small heat shock proteins (sHSPs) are members of the heat shock protein (HSP) family that play an important role in temperature stress, and heat shock factors (HSFs) are transcriptional activators that regulate HSP expression. *Cotesia chilonis*, the major endoparasitoid of *Chilo suppressalis*, modulates the *C. suppressalis* population in the field. In this study, we cloned and characterized two genes from *C.*
*chilonis*: the heat-induced HSP11.0 gene (*Cchsp11.0*) that consisted of a 306-bp ORF, and the master regulator HSF (*Cchsf*) containing an 1875-bp ORF. *Cc*HSP11.0 contained a chaperonin cpn10 signature motif that is conserved in other hymenopteran insects. *Cc*HSF is a typical HSF and contains a DNA-binding domain, two hydrophobic heptad repeat domains, and a C-terminal trans-activation domain. Neither *Cchsp11.0* or *Cchsf* contain introns. Real-time quantitative PCR revealed that *Cchsp11.0* and *Cchsf* were highly induced at 36 °C and 6 °C after a 2-h exposure. Overall, the induction of *Cchsf* was lower than *Cchsp11.0* at low temperatures, whereas the opposite was true at high temperatures. In conclusion, both *Cchsp11.0* and *Cchsf* are sensitive to high and low temperature stress, and the expression pattern of the two genes were positively correlated during temperature stress.

## 1. Introduction

*Chilo suppressalis* (Walker) is a detrimental rice pest that widely distributed in rice fields of China, which is known as the striped rice stem borer [1]. *Cotesia chilonis* (Munakata) is the major endoparasitoid of *Chilo suppressalis* (Walker), and has become the dominant parasitic wasp of *C. suppressalis* as global temperatures have risen [2,3,4].

Global warming has garnered widespread attention on the influence of temperature [5]. As an important environmental factor, temperature influences the growth and development of insects and the structure and function of cellular proteins [6,7,8]. Insects adopt various survival strategies when exposed to temperature stress; this heat shock response (HSR) is rapidly induced by heat or other environmental and physiological stressors [9]. The HSR can remove misfolded or damaged proteins in the cytoplasm and nucleus and also contributes to the expression of genes encoding heat shock proteins (HSPs). The HSPs are highly-conserved proteins that are induced by suboptimal temperatures; they function to prevent protein denaturation and help restore conformation and biological activity [2]. The main function of HSPs is to improve the tolerance of organisms to various environmental stresses, such as temperature, hunger, heavy metals, ultraviolet rays and pesticides; furthermore, and they can be used as biomarkers for various stresses [10,11]. On the basis of molecular weight and amino acid similarity, the HSP superfamily can be divided into HSP100, HSP90, HSP70, HSP60, HSP40 and small heat shock proteins (sHSPs) [12,13]. sHSPs are relatively small (about 12–43 kDa), and possess diverse amino acid sequences; however, they share a conserved α-crystallin domain (ACD) [14].

Heat shock transcription factors (HSFs) are crucial regulatory factors of the HSR that are conserved in eukaryotes [15,16]. HSFs utilize a conserved regulatory mechanism where heat shock transcription factor 1 (HSF1) is the major transcription factor to regulator; this factor is expressed in most tissues and cells in response to heat stress [17]. HSF1 is comprised of four conserved domains including a DNA-binding domain (DBD), hydrophobic heptad repeat domains-heptad repeat ofhydrophobic amino acids A and B (HR-A/B) and C-terminal heptad repeat (HR-C) and a C-terminal trans-activation domain (CTAD) [18]. The structure and function of HSF1 has been well-studied in mammals and *Drosophila melanogaster* (Meigen) [19,20].

sHSPs are a common feature of insects, and numerous studies have reported the response of insect sHSPs to temperature [21,22,23,24]. For example, *Sihsp19.6*, *Sihsp20.6* and *Sihsp21.4* in *Sesamia inferens* and *Lshsp19.5*, *Lshsp20.8*, and *Lshsp21.7* in *Liriomyza sativa* were up-regulated when exposed to low temperature stress [25,26]. In *Chilo suppressalis*, *Cshsp23.9* was induced at high temperature (36 °C) but did not respond to low temperature stress [27]. In *Plutella xylostella*, 12 *Pxhsps* were significantly induced by high and low temperatures [28]. In addition, a few reports exist documenting HSF1 in other insect species including *Bactrocera dorsalis* (Hendel), *Helicoverpa armigera* (Hübner), *Bombyx mori* (Linnaeus) and *Mamestra brassicae* (Linnaeus) [18,29,30]; however, studies showing a relationship between the expression of *hsfs* and *shsps* under temperature stress in insects are lacking.

Many studies demonstrated that various sHSPs play a significant part in thermotolerance of insects [1,2,27]. Our previous studies have shown that *Cc*HSPs play an important role in temperature tolerance; however, with the exception of *Cchsp40*, there is no evidence for the role of sHSPs in thermotolerance of *C. chilonis* [2,27]. However, Moreover, the regulatory mechanism between HSFs and sHSPs on protection is worthy of further study, which can start by studying the expression link between *hsfs* and *shsps in* response to temperature stress. In this study, a second gene encoding a sHSP, *Cchsp11.0*, and an HSF factor, *Cchsf*, were cloned and characterized in response to thermal stress. The results provide new insights into thermoregulation of *C. chilonis* in response to climate change.

## 2. Materials and Methods

### 2.1. Experimental Insects

*C. suppressalis* and *C. chilonis* were collected from a suburb of Yangzhou (32.39 °N, 119.42 °E) and reared under the laboratory conditions at 27 ± 1 °C, 60–70% RH and a 16:8 h (light/dark) photoperiod [2]. *C. suppressalis* larvae were reared on an artificial diet [1]. *C. chilonis* adults were supplied with a 10% honey/water solution and propagated using 5th instar larvae of *C. suppressalis* as hosts.

### 2.2. Sample Treatments

#### 2.2.1. Different Temperature Treatments

For different temperature treatments, one-day-old adults of *C. chilonis* were subjected to −13, −12, −9, −6, −3, 0, 27, 30, 33, or 36 °C for 1 h in a constant-temperature incubator [2]; samples were then placed in a climate-controlled incubator and allowed to recover at 27 °C for 1 h. Each treatment contained 30 one-day old adults, and all treatments were replicated three times. 

#### 2.2.2. Thermal Treatment at Different Times

For thermal treatment at different times, one-day-old adults of *C. chilonis* were subjected to 36 °C or −6 °C for 15 min, 30 min, 1 h, 2 h, 4 h, or 8 h in a controlled temperature incubator; samples were then transferred to a climate-controlled incubator to recover at 27 °C for 1 h. Controls were maintained at 27 °C for 1 h. Temperature selection for the treatments were based on previous articles [2]. Each treatment contained 40 one-day old adults, and all treatments were replicated four times.

### 2.3. Total RNA Isolation and Synthesis of First Strand cDNA

Total RNA was extracted from *C. chilonis* using the SV Total RNA Isolation System (Promega, Madison, WI, USA). RNA purity and concentration were measured by agarose gel electrophoresis and spectrophotometry (NanoDrop One, Thermo Fisher Scientific, Madison, WI, USA). The first strand of cDNA was synthesized using RevertAid First Strand cDNA Synthesis Kit (Thermo, Madison, WI, USA) and cDNAs for 5′- and 3′- RACE were synthesized by SMARTerTM cDNA Amplification Kit (Clontech, Mountain View, CA, USA).

### 2.4. Cloning and Genome Amplification

Partial gene sequences were obtained from the *C. chilonis* transcriptome (unpublished data), and according to the primer design principle and nucleotide sequence we obtained, specific primers were designed by Primer Premier 5 to verify fragments using the first strand of cDNA as template (Table 1). Full-length cDNA sequences of genes were obtained with 5′- and 3′-RACE (SMARTerTM RACE, Clontech, Mountain View, CA, USA), and gene-specific primers were designed for verifying full-length cDNA sequences using the 5′-RACE template (Table 1). Genomic DNA of *C. chilonis* adults was extracted using the AxyprepTM Multisource Genomic DNA Kit (Axygen, New York, NY, USA), and primers (Table 1) were designed to amplify genomic fragments of *Cchsp11.0* and *Cchsf* for subsequent cloning or sequencing.

### 2.5. Sequence Analysis of Genes

ORFs were identified with ORF Finder (https://www.ncbi.nlm.nih.gov/orffinder/) (accessed on 12 December 2020), and deduced amino acid sequences were aligned with Clustal X [32]. Sequence analysis tools on the ExPASy Molecular Biology Server including Translate, Compute pI/MW, and Blast (Swiss Institute of Bioinformatics, Lausanne, Switzerland), were used to analyze the predicted sequences. Motif Scan (https://prosite.expasy.org/) (accessed on 12 December 2020) and InterPro (http://www.ebi.ac.uk/interpro/) (accessed on 12 December 2020) were used to identify motifs characteristic of the sHSPs family. Amino acid sequences of 18 sHSPs and 23 HSFs were downloaded from NCBI (https://www.ncbi.nlm.nih.gov/) (accessed on 12 December 2020). Then phylogenetic trees were constructed by the neighbor-joining minimum evolution, maximum likelihood and maximum parsimony methods with 1000 bootstrap replicates using MEGA X [33].

### 2.6. Real-Time qPCR Analysis

Total RNA of different treatments was isolated as described above. The Bio-Rad iScriptTM cDNA Synthesis Kit (Bio-Rad, Laboratories, Berkeley, CA, USA) was used to reverse-transcribe 0.5 µg total RNA into first strand cDNA. The primers used for real-time quantitative PCR (Table 1) were designed according to the full-length cDNA sequence of genes. Real-time PCR reactions were conducted by using SYBR Green I in a 20 μL reaction volume containing 10 μL iTaqTM SYBR^®^ Green Supermix (Thermo, Madison, WI, USA), 6 μL ddH_2_0, 2 μL cDNA template and 1 μL 10 μM each of the corresponding forward and reverse primers. PCR conditions were as follows: 3 min initial denaturation step at 95 °C, followed by 40 cycles of 15 s denaturation at 95 °C, and 30 s annealing at the Tm for each gene (Table 1). Melting curve analysis was carried out to evaluate the homogeneity of amplified PCR products, and each PCR reaction was replicated in triplicate.

### 2.7. Statistical Analysis

Relative quantitative analysis was performed by the 2^−ΔΔCt^ method to obtain the relative expression level of each treatment. *H3* encoding histone 3 was regarded as a low-temperature reference gene, and *GAPDH* encoding glyceraldehyde-3-phosphate dehydrogenase was regarded as the high-temperature reference gene [31]. Differences in mean values were analyzed using one-way ANOVA. Homogeneity of variances among treatments was measured by Levene’s test, and significance differences were assessed by Tukey’s test. All statistics were performed using SPSS16.0 software and represented as means ± SE (standard error).

## 3. Results

### 3.1. Characteristics of Sequenced Genes

The full-length cDNA sequence of *Cchsp11.0* was 508 bp (GenBank accession no. MN176104) (https://www.ncbi.nlm.nih.gov/ accessed on 10 October 2020) and contained a 132-bp 5′ untranslated region (UTR), a 306-bp open reading frame (ORF), and 70-bp 3′ UTR (Appendix A). The predicted *Cc*HSP11.0 protein contained 101 amino acids with a molecular mass of 11.0 kDa and theoretical isoelectric point (*pI*) of 8.03. MotifScan indicated that *Cc*HSP11.0 contained a chaperonin cpn10 signature sequence (residues 8–32) but lacked an α-crystallin domain. Multiple sequence alignments showed a 91.90% sequence identity between *Cc*HSP11.0 and orthologous in other hymenopteran sHSPs (Figure 1A). The comparison of cDNA and genomic DNA of *Cchsp11.0* cDNA indicated that the absence of introns (Figure 2A).

Full-length cDNA of *Cchsf* was 2073 bp (GenBank accession no. MT157267) and contained a 95-bp 5′ UTR, an 1875-bp ORF, and a 103-bp 3′ UTR (Appendix A). The deduced protein product contained 624 amino acids with a predicted mass of 70.03 kDa and pI of 4.99. InterPro analysis indicated that *Cc*HSF contained four conversed domains including a DNA-binding domain (DBD), the hydrophobic heptad repeats HR-A/B and HR-C, and the C-terminal trans-activation domain (CTAD); these spanned residues 10–113, 131–209, 484–518, and 569–583, respectively. Multiple sequence alignments revealed that *Cc*HSF shared 31.21%, 55.83%, 30.81% and 30.97% identity with HSFs in *B. mori*, *Apis mellifera*, *M. brassicae*, and *H. armigera* (Figure 1B). No introns were found in *Cchsf* when cDNA and genomic sequences were compared (Figure 2B).

### 3.2. Phylogenetic Analysis of Genes

Similar phylogenetic trees were obtained using neighbor-joining, maximum likelihood, maximum parsimony and minimum evolution methods. The dendogram in Figure 3 shows the results obtained with the neighbor-joining method due to its relatively accurate and fast calculation speed. Analysis using Clustal X and MEGA X [33] indicated that CcHSP11.0 was closely related to other hymenopteran insects (Figure 3A); furthermore, it should also be noted that these orthologous proteins contain the chaperonin cpn10 signature. The deduced protein sequence of *Cc*HSF shared high similarity with other insects and select mammalian orthologues (Figure 3B) that also contained the four conserved HSF domains (data not shown).

### 3.3. Gene Expression in Response to Different Temperatures

The relative mRNA levels of *Cchsp11.0* and *Cchsf* were monitored at temperature gradients ranging from −13 °C to 36 °C (Figure 4). Expression of *Cchsp11.0* and *Cchsf* showed similar expression patterns at different temperatures. The expressions of these two genes were both up-regulated by cold stress while remained unchanged by heat stress (*Cchsp11.0*, *F*_9,20_ = 31.933, *P* < 0.001; *Cchsf*: *F*_9,19_ = 63.093, *p* < 0.001). Compared to the control (27 °C), the relative expression of *Cchsp11.0* and *Cchsf* were remarkably up-regulated at −6 °C, and the expression was 12.33-fold and 65.45-fold higher than the control, respectively.

### 3.4. Time Course of Gene Expression After Temperature Treatments

The relative mRNA levels of *Cchsp11.0* and *Cchsf* were monitored using thermal treatments at different intervals (15 min to 8 h); however, samples monitored at the 8 h time point were discarded due to high mortality. Gene expression patterns for *Cchsp11.0* and *Cchsf* were positively correlated for the different intervals after exposure to 36 °C (Figure 5). The expression of *Cchsp11.0* and *Cchsf* were both up-regulated at 15 min and 2 h after exposure to 36 °C (Figure 5), and expression was 10.82- and 6.47-fold higher than the control at the 2-h time interval, respectively. At −6 °C, both *Cchsp11.0* and *Cchsf* were significantly up-regulated at the 1 and 2 h intervals as compared to the control (27 °C); expression at the 4 h time period was negligible. The greatest expression of *Cchsp11.0* and *Cchsf* was observed at 2 h; at this time point, expression levels were 7.61- and 43.01-fold higher than control, respectively (Figure 6).

## 4. Discussion

HSPs and HSFs are key regulators and effectors of the heat shock response. In this study, we cloned and characterized the heat-induced *Cchsp11.0* and the master regulator *Cchsf* in *C. chilonis*. Multiple sequence alignments and phylogenetic analysis showed that *Cc*HSP11.0 and *Cc*HSF are highly conserved and closely related to orthologues in other hymenopteran insects. *Cc*HSF contained four conversed domains widely found in heat shock factors. *Cc*HSP11.0 contained a chaperonin cpn10 signature, which is not typical of sHSPs but belongs to a major of HSPs and controversially known as sHSP family [34,35]. The 10 kDa heat shock protein, serving as molecular chaperone, co-chaperones with the HSP60 [36]. Further verification of gene sequences and genome structure revealed that there were no introns in either *Cchsp11.0* or *Cchsf,* which may be part of the strategy used by *C. chilonis* to quickly activate transcription of these genes in response to temperature stress.

Temperature is a critical environmental factor that affects insect growth, development, distribution and abundance [37]. When subjected to high or low temperature stress, insects may adopt different coping strategies, such as avoidance behaviors or changing physiological functions to tolerate temperature stress [38]. Previous studies have shown that the tolerance of insects to temperature extremes is largely due to *hsp* regulation and changes in *hsp* expression [9,12]. In this study, *Cchsp11.0* expression was significantly up-regulated at −6 °C; however, *Cchsp11.0* did not respond to high temperature stress. In *C. suppressalis*, *Cshsp21.5* was up-regulated by low but not high temperatures [23,39]; whereas, *Cshsp21.4* and *Cshsp21.7a* were insensitive to temperature [38]. The expression levels of four *shsps* in *Laodelphgax striatellus* were up-regulated by high temperature [40], and *Lthsp20.0* in *L. trifolii* was induced by both high and low temperatures [28]. These results indicate that insects have a complex network of small heat shock proteins with diverse expression patterns. The temperature tolerance of insects to extreme temperatures may be potentially improved by the interaction of different sHSPs; therefore, the expression patterns of *shsps* in *C. chilonis* warrant further study.

Thermotolerance in organisms is generally accompanied by the induction of HSPs, which are regulated by heat shock transcription factors. In eukaryotic cells exposed to high temperatures, HSF1 was shown to activate its own transcription [16]. In this study, expression of *Cchsf* was not induced in response to high temperatures, but did show increased expression during cold stress. These results are consistent with findings reported by Guo (2013) who found that *Bthsf* in *Bemisia tabaci* can be induced by low but not high temperatures [41]. In contrast, *hsf* in *Marsupenaeus japonicus* was induced in response to heat stress [42], which suggests that *hsf* expression patterns vary in different insect species. In general, there are relatively few studies on the expression pattern of insect HSFs during temperature stress.

In time course experiments, the expression patterns of *Cchsp11.0* and *Cchsf* transcription were positively correlated. Both *Cchsp11.0* and *Cchsf* were induced at 15 min and 2 h after exposure to 36 °C. In addition to providing protection from heat stress, HSP10 functions as a chaperone to prevent the irreversible aggregation of proteins and as a co-chaperonin with HSP60 [35]. *Cchsp11.0* and *Cchsf* expression increased at the 15 min time point in response to heat stress and then declined until another round of induction 2 h. The decreased expression of *Cchsp11.0* at 30 min and 1 h may be related to the regulation of other HSPs [2], and the decline in *Cchsf* expression at these times may be related to the negative regulation of HSF by HSP. Previous studies showed that many small heat shock proteins have their maximum expression after 2 h of temperature treatment, such as *Cshsp702*, *Cshsp19.8*, *Cshsp21.7b*, *Cshsp21.5*, which is consistent with our experimental results [1,23]. At 4 h, the expression of *Cchsf* decreased again, suggesting that HSF transcriptional activity is weakened somewhat over time [19,43]. Overall, *Cchsf* transcription was induced at lower levels than *Cchsp11.0*, perhaps because the regulation involves a conversion that affects its activity, such as its conversion from a monomer to a multimeric form or phosphorylation [41,44]. Additionally, the transcription of both *Cchsp11.0* and *Cchsf* was induced at −6 °C and was highest at 2 h; however, expression of the two genes was minimal at 4 h.

## 5. Conclusions

In conclusion, both *Cchsp11.0* and *Cchsf* are sensitive to high and low temperature stress in *C. chilonis*, with maximal expression at 36 °C and 6 °C after 2 h. The expression pattern of the two genes were strongly correlated at different times; however, the underlying mechanisms warrant further study in order to effectively control *C. suppressalis* using *C. chilonis*.

## Figures and Tables

**Figure 1 insects-12-00322-f001:**
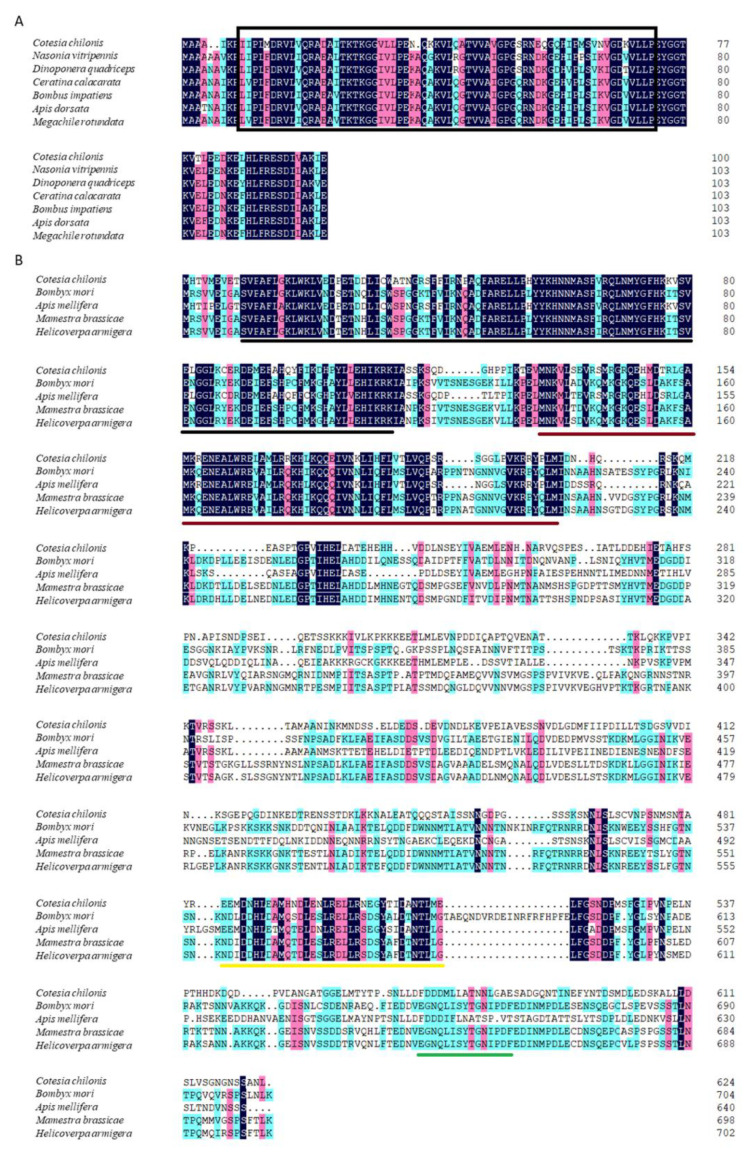
Amino acid sequence alignment of *Cc*HSP11.0 (**A**) and *Cc*HSF (**B**) from *C. chilonis* with orthologous proteins in *B. mori*, *A. mellifera*, *M. brassicae*, and *H. armigera*. Identical amino acids are shaded with the same color. The chaperonin cpn10 signature sequence is marked by a rectangle. The DNA-binding (DBD) motif, hydrophobic heptad repeats-heptad repeat ofhydrophobic amino acids A and B (HR-A/B) and C-terminal heptad repeat (HR-C) and C-terminal transactivation domain (CTAD) are underscored in black, red, yellow and green, respectively. Accession numbers of species are noted in Appendix A.

**Figure 2 insects-12-00322-f002:**
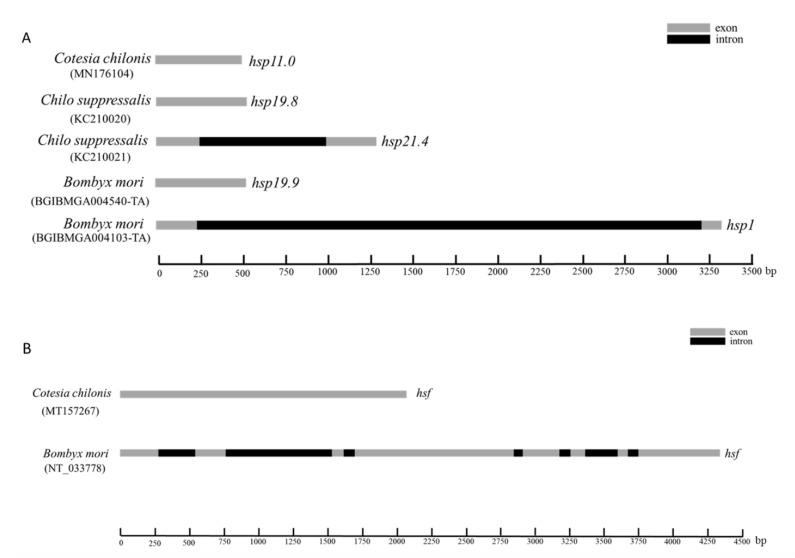
Diagrammatic representation of the genomic forms of *shsp* and *hsf* genes in *C. chilonis*, *C. suppressalis* and *B. mori*. (**A**) Gene structure of *hsp11.0* in *C. chilonis*, *hsp19.8* and *hsp.21.4* in *C. suppressalis* and *hsp19.9* and *hsp1* in *B. mori*. (**B**) Structure of *hsf* genes in *C. chilonis* and *B. mori*. Gray and black rectangles are used to denote exons and introns, respectively.

**Figure 3 insects-12-00322-f003:**
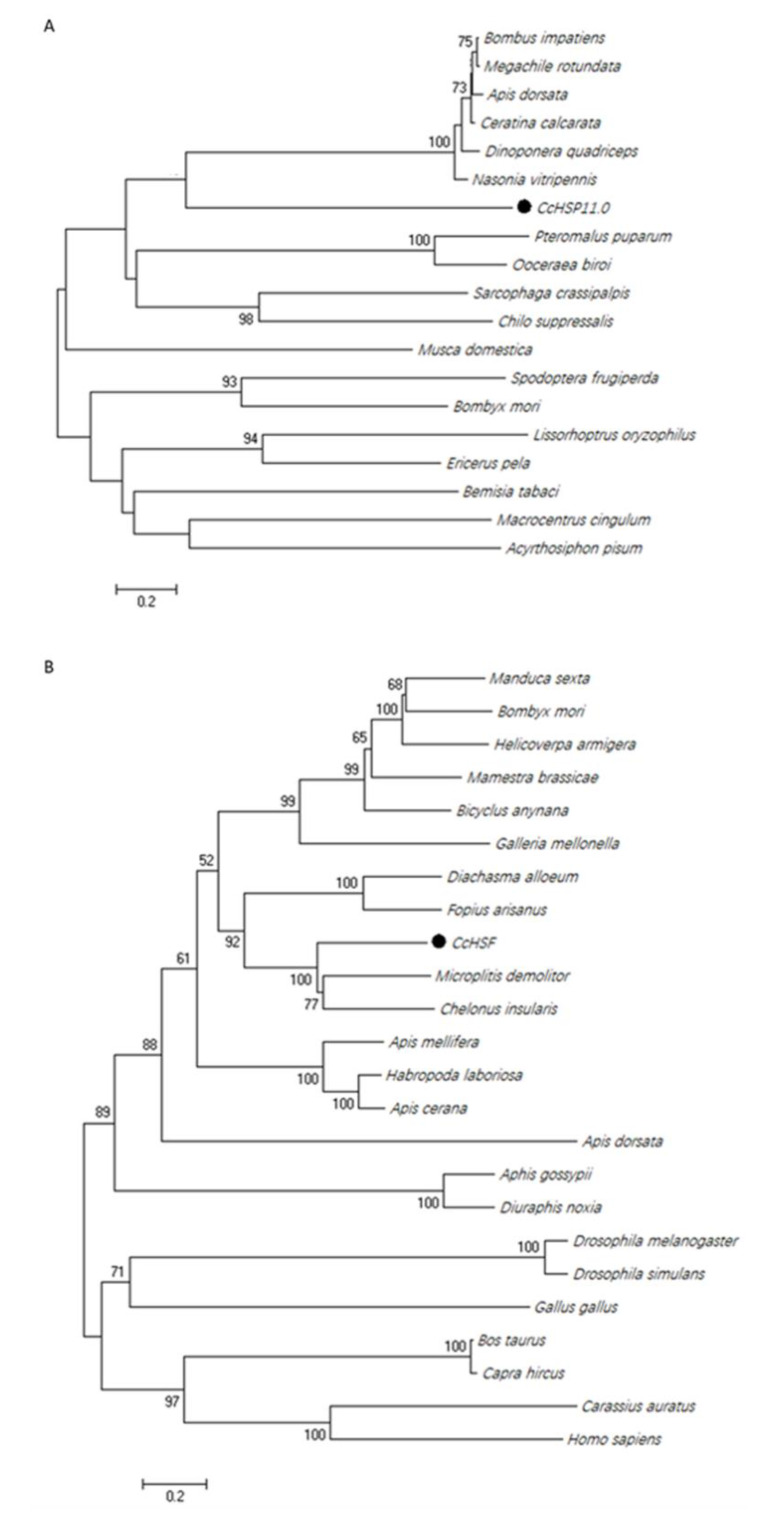
Phylogenetic analysis of HSP11.0 and heat shock transcription factor (HSF) proteins. (**A**) Relatedness of *Cc*HSP in *C. chilonis* to orthologues in other insects. (**B**) relatedness of *Cc*HSF to proteins in other insects and mammals. Trees were generated with MEGA X, and solid circles indicate the location of CcHSP11.0 and *Cc*HSF. Numbers on the branches are bootstrap values (1000 replicates), and only bootstrap *p* values > 50 are shown. Accession numbers are provided in Appendix A.

**Figure 4 insects-12-00322-f004:**
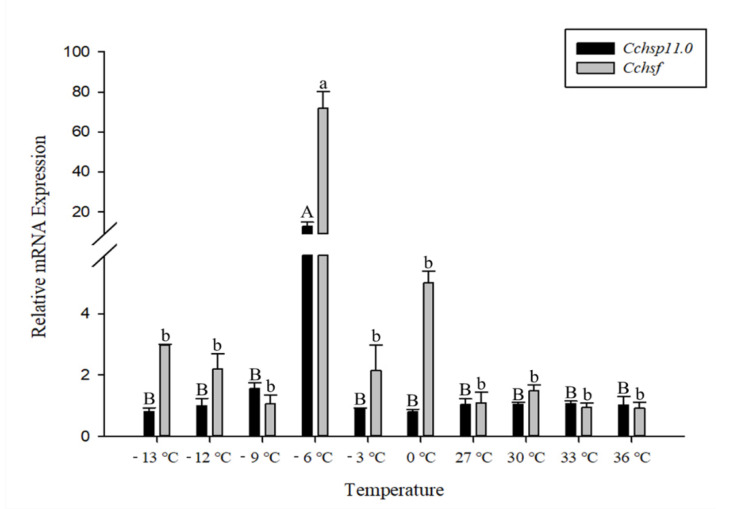
Relative mRNA expression levels of *Cchsp11.0* and *Cchsf* at different temperatures. The control treatment is 27 °C. Statistics are presented as means ± SE (standard error). Columns labeled with different letters indicate significance using one-way ANOVA followed by Tukey’s multiple comparison analysis (*p* < 0.05). Uppercase letters indicate the significance of *Cchsp11.0* and lowercase letters indicate the significance of *Cchsf*.

**Figure 5 insects-12-00322-f005:**
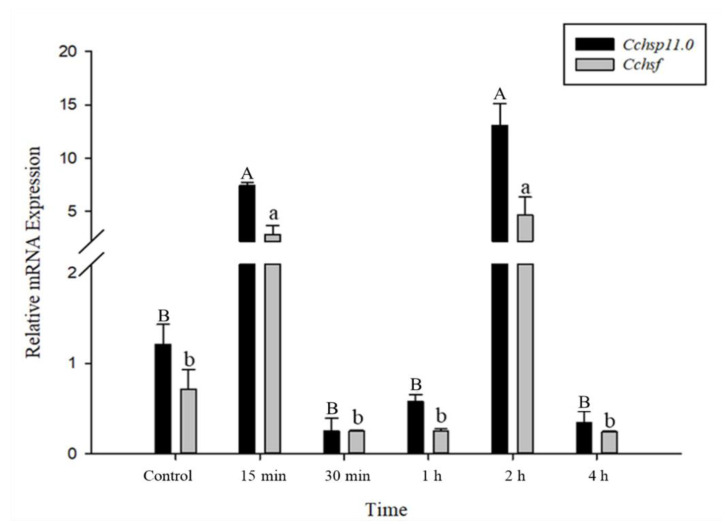
Relative mRNA expression levels of *Cchsp11.0* and *Cchsf* at different intervals after exposure to 36 °C. Statistics represent means ± SE. Columns labeled with different letters indicate significant differences between times using one-way ANOVA followed by Tukey’s multiple comparison analysis (*p* < 0.05). Uppercase letters indicate the significance of *Cchsp11.0* and lowercase letters indicate the significance of *Cchsf*.

**Figure 6 insects-12-00322-f006:**
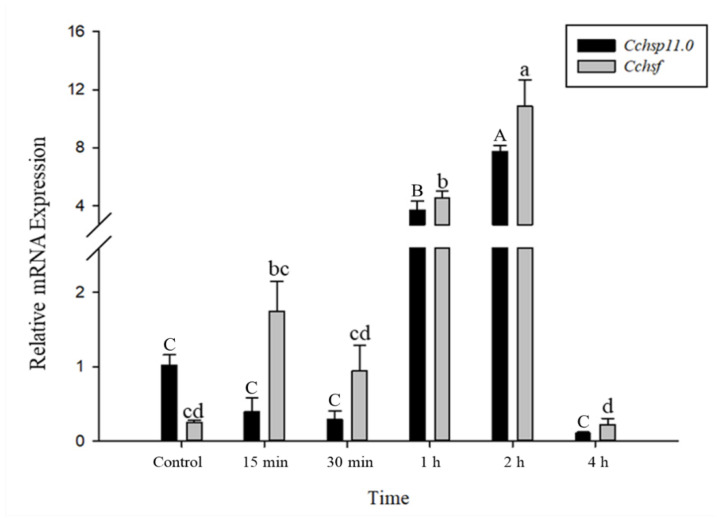
Relative mRNA expression levels of *Cchsp11.0* and *Cchsf* at different intervals after exposure to −6 °C. Statistics are represented as means ± SE. Columns labeled with different letters indicate significant differences between times using one-way ANOVA followed by Tukey’s multiple comparison analysis (*p* < 0.05). Uppercase letters indicate the significance of *Cchsp11.0* and lowercase letters indicate the significance of *Cchsf*.

**Table 1 insects-12-00322-t001:** Primers used in this study.

Primer Name	Primer Sequences (5′-3′)	Tm (°C)	ε ^b^ (%)	R^2^ ^c^
Fragment verification				
*hsp11.0-F*	CGGGAACAAATCAACAT	46.3		
*hsp11.0-R*	ACTCGGTCCATCAAAGG	51.3		
*hsf-F*	AGAACGCAACAACCAAG	50.0		
*hsf-R*	CAACTACAGAACCATCAGAG	45.0		
Rapid-amplification of cDNA ends (RACE)				
*hsp11.0-5′*	GAGCCAGGTCCAACAGCAACTACAG	62.4		
*hsp11.0-3′*	CGTTCAAAGAGCCGATGCTATAA	54.8		
*hsf-F*	CTTGCTGCTGGAGCCTGGATCAC	63.5		
*hsf-R*	ATTCCAGACATCCTACTCACCTC	55.7		
Verification of full-length cDNA				
*hsp11.0-F*	AGTTATTCACCAGCAACGT	51.1		
*hsp11.0-R*	GTTTGATAATTTCATAGAGC	42.4		
*hsf-F*	ATCACTAATACGACTCACTATAGGG	52.5		
*hsf-R*	TTTGTTTATAGTACGCAAGTCG	51.8		
Verification of genome				
*hsp11.0-F*	CTCAGATCTTATTCTTTCAT	42.6		
*hsp11.0-R*	GTTTGATAATTTCATAGAGC	42.4		
*hsf-F*	ATCACTAATACGACTCACTATAGGG	52.5		
*hsf-R*	GAGCTGAATAAATACACTCACCA	51.8		
Real-time quantitative PCR ^a^				
*hsp11.0-F*	ACAAAGTTCTCCTCCCCG	59.4	90.0	0.988
*hsp11.0-R*	GCAACAATGTCTGATTCACG			
*hsf-F*	TTAGGTGCTGAAAGTGCCGA	60.0	117.3	0.904
*hsf-R*	AGTACGCAAGTCGAGCTGAA			
Reference gene in qRT-PCR ^a^				
*H3-F*	CGTCGCTCTTCGTGAAATCA	58.1	97.4	0.978
*H3-R*	TCTGGAAACGCAAGTCGGTC			
*GAPDH-F*	GAAGGTGGTGCCAAGAAAG	54.0	106.7	0.978
*GAPDH-R*	GCATGGACAGTGGTCATAAGA			

**Note:**^a^ The qPCR primers used in this study were validated [31]. ^b^ Real-time qPCR efficiency (calculated from the standard curve). ^c^ Coefficient of determination.

## Data Availability

Data are contained within the article or Appendix A.

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
