# Peer review of "Molecular Characterization of Heat-Induced HSP11.0 and Master-Regulator HSF from Cotesia chilonis and Their Consistent Response to Heat Stress"

_insects, 2021, doi:10.3390/insects12040322_

Round 1

Reviewer 1 Report

Fu-Jing He and coworkers presents here the molecular characterisation of a small heat shock protein (CcHSP11.0) and its transcriptional activator (Cchsf) of the braconid parasitoid Cotesia chilonis. This parasitoid is of great importance for the biological control of Chilo suppressalis, a key pest of rice in East Asia, India and Indonesia, which was introduced in some European countries. The authors justify the interest of the work as per the distribution of the parasitoid species which seems highly driven by climate change. 

The work relies in the molecular characterisation of the two proteins by cDNA and genomic DNA sequencing and comparison towards other characterised sequences from other insects. One special part of the characterisation include temporal expression profiling after different temperature regimes. 

I consider that the work will be of great interest if authors perform some mayor changes, being the most important an in deep grammar and style usage.

The background section is a bit unorganised. I will suggest authors to rearrange the background first giving importance to the pest species and the target species, the parasitoid, and the climate change. Then focus on the implication of heat shock proteins in termal tolerance. Ending with how are these proteins organised in a well orchestrated play from protecting cellular proteins to protection of the whole insect. 

At the M&M section, please indicate the age of used C. chilonis in point 2.2. Clarify what are the treatments, ie. splitting it into 2.2.1 for thermal tolerance and 2.2.2 for time under each of the two tested critical temperatures (high 36 and low -6C). In section 2.3, please add kit used for cDNA synthesis. In section 2.4, please add rationale of how primers were designed. In page 3 you indicated partial gene sequences (section 2.4) were obtained from the C. chilonis transcriptome. Which transcriptome? please add the corresponding reference (ie. Qi et al 2015 Arch Insect Biochem Physiol 88(4): 203-221; or the appropriate one ???) and indicate how you get these primer sequences. As presented primers in Table 1 do not contain bubbles, should be sHSP or HSFs universal ones?? I can imagine than based on cDNA sequences, primers for genomic DNA were designed. 

In point 2.5, please specify why you used MEGA7 instead the actual version MEGAX, and which nucleotide or protein distance model you used for each phylogenetic tree inference. 

In point 2.6, add the corresponding concentration of qPCR primers. There is no Table 2, the Tm of each primer is located in Table 1. 

In point 2.7, the relative 2-DDCt method (sorry, D accounts for Delta greek symbol) should be specified against each reference gene.

At Results section, please check duplicated information. In addition, in Figure 2, please indicate what are the values for each scale. I understand each scale is in base-pairs, isn't it?, but it should be indicated.  In section 3.2. Phylogenetic analysis of genes and Figure 3, you indicate bootstrap values > are shown, but in Fig 3A, you have branches, specially the one locating CcHSP11.0 with bootstrap values <50 (i.e. 35, 43, 49). rEmove these, and justify why you show NJ tree and not the more complex ones? At the Fig.3 legend you indicate there are protein trees, but is not stated at the corresponding M&M section, as I asked before. 

In section 3.3., would it be possible to merge Figure 4 and Figure 5?, is a single experiment, and a single statistical analysis for each protein. I thing it will be more useful for comparisons that both graphs are together. From these results, seems that the sHSP11.0 responds early in relative low temperatures (-6C), and then is stabilised. Might be meaning is a cold shock protein?

In section 3.4, should I assume CK (in Figs 6 and 7 are controls, isn't it?

At the attached file are some suggestions for English usage, but, the whole document should be revised. Introduction and discussion, will benefit from a rewording. 

Author Response

Dear Editor and Reviewers,

Submitted is our revised manuscript (insects-1136109, Molecular characterization of heat-induced HSP11.0 and master-regulator HSF from Cotesia chilonis and their consistent response to heat stress. According to reviewers’ suggestions, it has been improved seriously. Responses to your comments and suggestions by point-to- point are marked in blue color. I hope the revised version met the requirements of reviewers and also eased their concerns. I greatly appreciate the time and efforts by you and reviewers in reading and evaluating our manuscript and believe their suggestions to further improve our manuscript.

Sincerely,

Fu-Jing He, Yu-Zhou Du

Reviewer 2 Report

Small heat shock proteins (sHSPs) are members of the HSP family that play an important role in thermal stress, and heat shock factors (HSFs) are crucial regulatory factors of the HSR. In this study, the authors provided the information of  Cchsp11.0 and Cchsf . They found both Cchsp11.0 and Cchsf are sensitive to high and low temperature stress. The expression pattern of the two genes were positively correlated during temperature stress. The significance and innovation point of the study are not clear in the manuscript. And there are 6 main issues with the text:

1. In the Introduction section, the authors raised a question “studies showing a relationship between HSFs and sHSPs in insects are lacking”. However, the conclusion in this study only revealed that the expression pattern of the two genes were positively correlated during temperature stress, which not thoroughly address the question in the introduction. The authors should have a clear scientific question or hypothesis.

2. The authors need to provide more information to explain the reason of studying Cchsp11.0 in the introduction.

3. In the Materials and Methods 2.6, although the qPCR primers used in this study were validated, the authors should supplement the amplification efficiency and R2 of all qPCR primers.

4. In the Results of Figure 4, 5, 6 and 7, it is better for the authors to indicate the statistical analysis results of two genes with uppercase and lowercase letters.

5. Since Figure 4 has shown that expression of Cchsp11.0 and Cchsf remained unchanged at increasingly elevated. I can't understand why authors also carry out the experiment of Figure 6. Meanwhile, in Ffigure 6, the expression of Cchsp11.0 and Cchsf were both up-regulated at 15 min and 2 h after exposure to 36℃. Has the author considered to carry out the experiment of Figure 4 again at 15 min or 2 h?

6. The authors should fully explain and discuss why the increased expression of Cchsp11.0 and Cchsf at 2 h in the Discussion section. According to the current way of writing, I feel that the authors have not explained it clearly.

Author Response

(The authors gave the same response as above.)

Round 2

Reviewer 1 Report

Dear authors, 

thank you for following all my comments to your manuscript. 

I consider the actual version is almost ready for acceptance, but, I have detected some small things that should be fixed before. 

Maybe is something related with your reference manager software and the new manuscript version. As references 30 to 35, and 38 to 40 have disappeared from the list of references. New references should be edited to match Insect journal style.

And the only last point to be checked is again Table 1.  You have added two new columns, Eb and R2, and added a 'Note' below the table stating "The qPCR primers used in this study were validated" (this table footnote should be called within the table). Should I assume that Eb accounts for Efficiency and R2 for linearity? Please clarify this point, and make the appropriate modifications. Efficiency is usually indicated by the greek letter Epsilon (ε) without any super index, unless the super index is used to make the description as a Table footnote, or correspond to one of the methods to determine efficiency.

Author Response

Submitted is our revised manuscript (insects-1136109, Molecular characterization of heat-induced HSP11.0 and master-regulator HSF from Cotesia chilonis and their consistent response to heat stress. Thank you for your suggestions to my manuscript. Our revised manuscript has been improved again. About the questions you raised, I will make the following remarks: 1. About references, it may be my negligence that caused your confusion. At present, we have adjusted the correct order and format. 2. The relevant parts of Table 1 have been revised and checked, see Table 1 and Line 131-133. I hope the revised version met the requirements of reviewers and also eased their concerns. I greatly appreciate the time and efforts by you and reviewers in reading and evaluating our manuscript and believe their suggestions to further improve our manuscript.

Sincerely,

Fu-Jing He, Yu-Zhou Du
